# Deep Learning Models of the Retinal Response to Natural Scenes

**Lane T. McIntosh**[*1], **Niru Maheswaranathan**[*1], **Aran Nayebi**[1],
**Surya Ganguli**[2,3], **Stephen A. Baccus**[3]
[1]Neurosciences PhD Program, [2]Department of Applied Physics, [3]Neurobiology Department
Stanford University
{lmcintosh, nirum, anayebi, sganguli, baccus}@stanford.edu

## Abstract

A central challenge in sensory neuroscience is to understand neural computations
and circuit mechanisms that underlie the encoding of ethologically relevant, natu-
ral stimuli. In multilayered neural circuits, nonlinear processes such as synaptic
transmission and spiking dynamics present a significant obstacle to the creation of
accurate computational models of responses to natural stimuli. Here we demon-
strate that deep convolutional neural networks (CNNs) capture retinal responses to
natural scenes nearly to within the variability of a cell's response, and are markedly
more accurate than linear-nonlinear (LN) models and Generalized Linear Mod-
els (GLMs). Moreover, we find two additional surprising properties of CNNs:
they are less susceptible to overfitting than their LN counterparts when trained
on small amounts of data, and generalize better when tested on stimuli drawn
from a different distribution (e.g. between natural scenes and white noise). An
examination of the learned CNNs reveals several properties. First, a richer set
of feature maps is necessary for predicting the responses to natural scenes com-
pared to white noise. Second, temporally precise responses to slowly varying
inputs originate from feedforward inhibition, similar to known retinal mechanisms.
Third, the injection of latent noise sources in intermediate layers enables our model
to capture the sub-Poisson spiking variability observed in retinal ganglion cells.
Fourth, augmenting our CNNs with recurrent lateral connections enables them to
capture contrast adaptation as an emergent property of accurately describing retinal
responses to natural scenes. These methods can be readily generalized to other
sensory modalities and stimulus ensembles. Overall, this work demonstrates that
CNNs not only accurately capture sensory circuit responses to natural scenes, but
also can yield information about the circuit's internal structure and function.

## 1 Introduction

A fundamental goal of sensory neuroscience involves building accurate neural encoding models that
predict the response of a sensory area to a stimulus of interest. These models have been used to shed
light on circuit computations [1, 2, 3, 4], uncover novel mechanisms [5, 6], highlight gaps in our
understanding [7], and quantify theoretical predictions [8, 9].

A commonly used model for retinal responses is a linear-nonlinear (LN) model that combines a linear
spatiotemporal filter with a single static nonlinearity. Although LN models have been used to describe
responses to artificial stimuli such as spatiotemporal white noise [10, 2], they fail to generalize to
natural stimuli [7]. Furthermore, the white noise stimuli used in previous studies are often low
resolution or spatially uniform and therefore fail to differentially activate nonlinear subunits in the

---

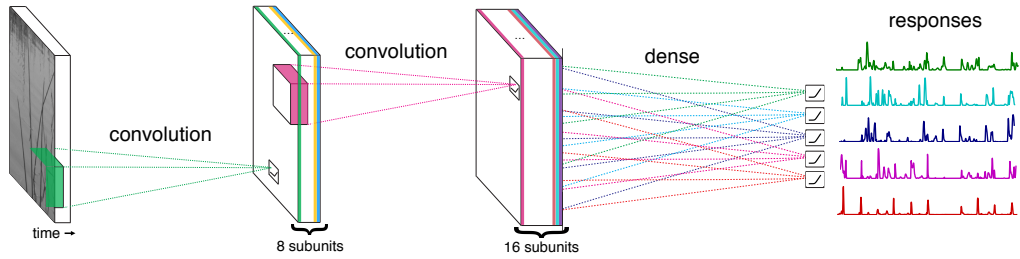

Figure 1: A schematic of the model architecture. The stimulus was convolved with 8 learned spatiotemporal filters whose activations were rectified. The second convolutional layer then projected the activity of these subunits through spatial filters onto 16 subunit types, whose activity was linearly combined and passed through a final soft rectifying nonlinearity to yield the predicted response.

retina, potentially simplifying the retinal response to such stimuli [11, 12, 2, 10, 13]. In contrast to the perceived linearity of the retinal response to coarse stimuli, the retina performs a wide variety of nonlinear computations including object motion detection [6], adaptation to complex spatiotemporal patterns [14], encoding spatial structure as spike latency [15], and anticipation of periodic stimuli [16], to name a few. However it is unclear what role these nonlinear computational mechanisms have in generating responses to more general natural stimuli.

To better understand the visual code for natural stimuli, we modeled retinal responses to natural image sequences with convolutional neural networks (CNNs). CNNs have been successful at many pattern recognition and function approximation tasks [17]. In addition, these models cascade multiple layers of spatiotemporal filtering and rectification–exactly the elementary computational building blocks thought to underlie complex functional responses of sensory circuits. Previous work utilized CNNs to gain insight into the neural computations of inferotemporal cortex [18], but these models have not been applied to early sensory areas where knowledge of neural circuitry can provide important validation for such models.

We find that deep neural network models markedly outperform previous models in predicting retinal responses both for white noise and natural scenes. Moreover, these models generalize better to unseen stimulus classes, and learn internal features consistent with known retinal properties, including sub-Poisson variability, feedforward inhibition, and contrast adaptation. Our findings indicate that CNNs can reveal both neural computations and mechanisms within a multilayered neural circuit under natural stimulation.

## 2 Methods

The spiking activity of a population of tiger salamander retinal ganglion cells was recorded in response to both sequences of natural images jittered with the statistics of eye movements and high resolution spatiotemporal white noise. Convolutional neural networks were trained to predict ganglion cell responses to each stimulus class, simultaneously for all cells in the recorded population of a given retina. For a comparison baseline, we also trained linear-nonlinear models [19] and generalized linear models (GLMs) with spike history feedback [2]. More details on the stimuli, retinal recordings, experimental structure, and division of data for training, validation, and testing are given in the Supplemental Material.

### 2.1 Architecture and optimization

The convolutional neural network architecture is shown in Figure 2.1. Model parameters were optimized to minimize a loss function corresponding to the negative log-likelihood under Poisson spike generation. Optimization was performed using ADAM [20] via the Keras and Theano software libraries [21]. The networks were regularized with an $\ell_2$ weight penalty at each layer and an $\ell_1$ activity penalty at the final layer, which helped maintain a baseline firing rate near 0 Hz.

We explored a variety of architectures for the CNN, varying the number of layers, number of filters per layer, the type of layer (convolutional or dense), and the size of the convolutional filters. Increasing the number of layers increased prediction accuracy on held-out data up to three layers, after which performance saturated. One implication of this architecture search is that LN-LN cascade models – which are equivalent to a 2-layer CNN – would also underperform 3 layer CNN models.

Contrary to the increasingly small filter sizes used by many state-of-the-art object recognition networks, our networks had better performance using filter sizes in excess of 15x15 checkers. Models were trained over the course of 100 epochs, with early-stopping guided by a validation set. See Supplementary Materials for details on the baseline models we used for comparison.

## 3 Results

We found that convolutional neural networks were substantially better at predicting retinal responses than either linear-nonlinear (LN) models or generalized linear models (GLMs) on both white noise and natural scene stimuli (Figure 2).

### 3.1 Performance

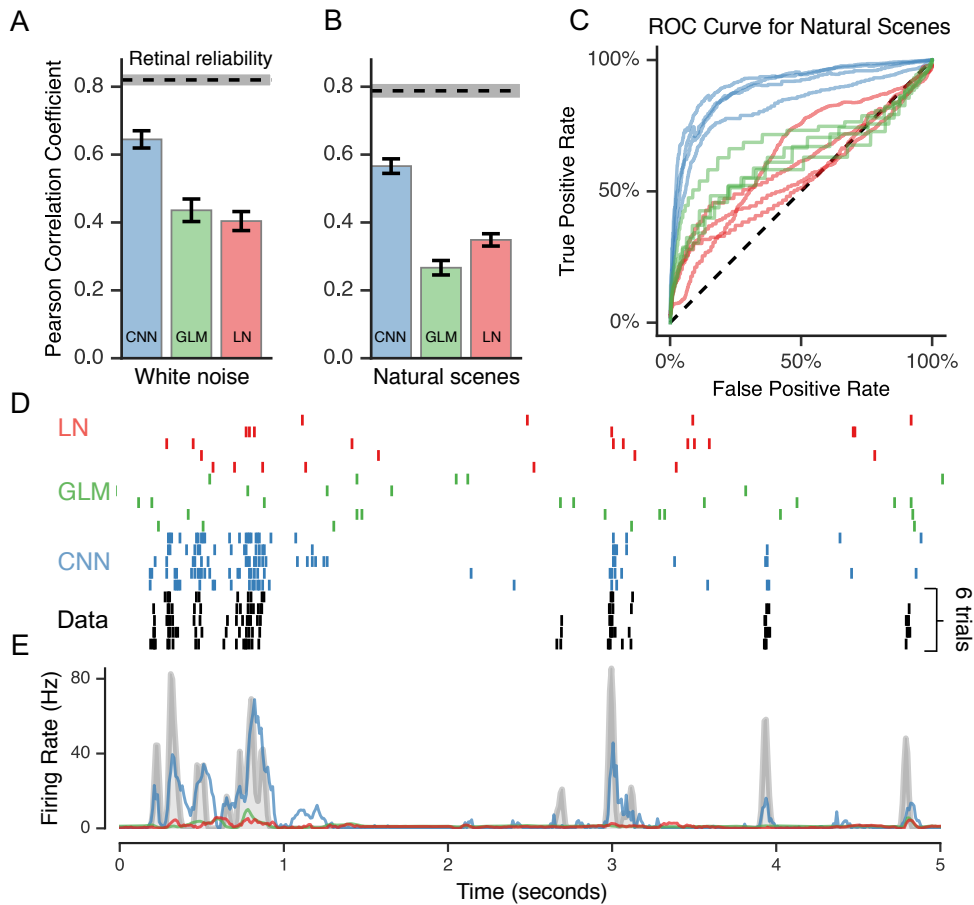

Figure 2: Model performance. (A,B) Correlation coefficients between the data and CNN, GLM or LN models for white noise and natural scenes. Dotted line indicates a measure of retinal reliability (See Methods). (C) Receiver Operating Characteristic (ROC) curve for spike events for CNN, GLM and LN models. (D) Spike rasters of one example retinal ganglion cell responding to 6 repeated trials of the same randomly selected segment of the natural scenes stimulus (black) compared to the predictions of the LN (red), GLM (green), or CNN (blue) model with Poisson spike generation used to generate model rasters. (E) Peristimulus time histogram (PSTH) of the spike rasters in (D).

LN models and GLMs failed to capture retinal responses to natural scenes (Figure 2B) consistent with previous results [7]. In addition, we also found that LN models only captured a small fraction of the response to high resolution spatiotemporal white noise, presumably because of the finer resolution that were used (Figure 2A). In contrast, CNNs approach the reliability of the retina for both white noise and natural scenes. Using other metrics, including fraction of explained variance, log-likelihood, and mean squared error, CNNs showed a robust gain in performance over previously described sensory encoding models.

We investigated how model performance varied as a function of training data, and found that LN models were more susceptible to overfitting than CNNs, despite having fewer parameters (Figure 4A). In particular, a CNN model trained using just 25 minutes of data had better held out performance than an LN model fit using the full 60 minute recording. We expect that both depth and convolutional filters act as implicit regularizers for CNN models, thereby increasing generalization performance.

## 3.2 CNN model parameters

Figure 3 shows a visualization of the model parameters learned when a convolutional network is trained to predict responses to either white noise or natural scenes. We visualized the average feature represented by a model unit by computing a response-weighted average for that unit. Models trained on white noise learned first layer features with small ($\sim$200 $\mu m$) receptive field widths (top left box in Figure 3), whereas the natural scene model learns spatiotemporal filters with overall lower spatial and temporal frequencies. This is likely in part due to the abundance of low spatial frequencies present in natural images [22]. We see a greater diversity of spatiotemporal features in the second layer receptive fields compared to the first (bottom panels in Figure 3). Additionally, we see more diversity in models trained on natural scenes, compared to white noise.

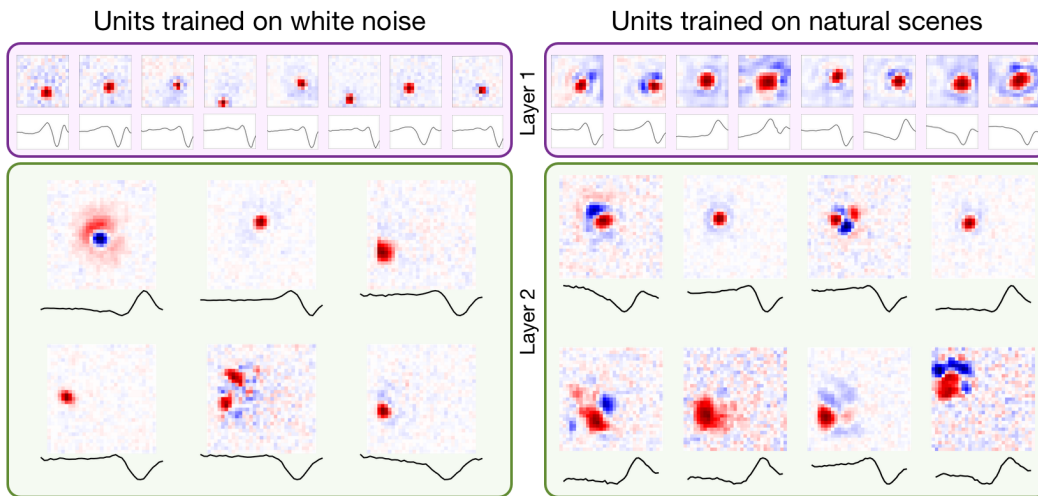

Figure 3: Model parameters visualized by computing a response-weighted average for different model units, computed for models trained on spatiotemporal white noise stimuli (left) or natural image sequences (right). Top panel (purple box): visualization of units in the first layer. Each 3D spatiotemporal receptive field is displayed via a rank-one decomposition consisting of a spatial filter (top) and temporal kernel (black traces, bottom). Bottom panel (green box): receptive fields for the second layer units, again visualized using a rank-one decomposition. Natural scenes models required more active second layer units, displaying a greater diversity of spatiotemporal features. Receptive fields are cropped to the region of space where the subunits have non-zero sensitivity.

## 3.3 Generalization across stimulus distributions

Historically, much of our understanding of the retina comes from fitting models to responses to artificial stimuli and then generalizing this understanding to cases where the stimulus distribution is more natural. Due to the large difference between artificial and natural stimulus distributions, it is unknown what retinal encoding properties generalize to a new stimulus.

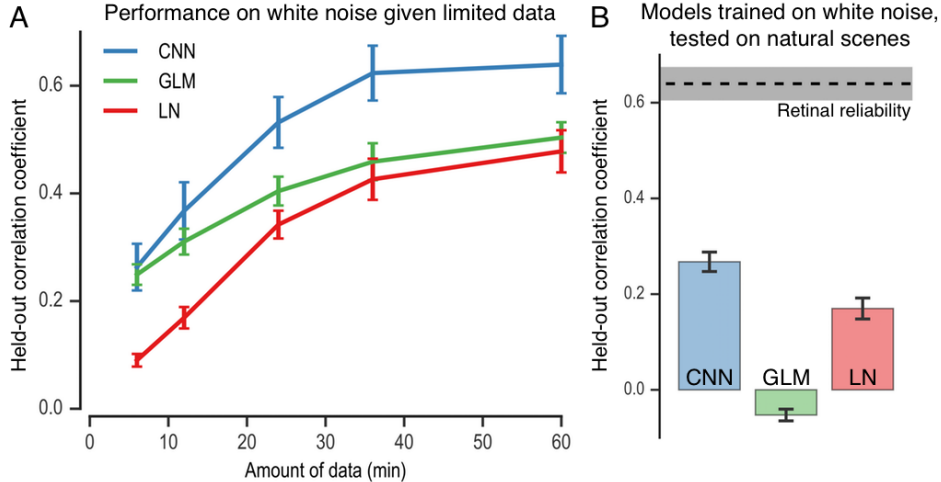

Figure 4: CNNs overfit less and generalize better across stimulus class as compared to simpler models. (A) Held-out performance curves for CNN (∼150,000 parameters) and GLM/LN models cropped around the cell's receptive field (∼4,000 parameters) as a function of the amount of training data. (B) Correlation coefficients between responses to natural scenes and models trained on white noise but tested on natural scenes. See text for discussion.

We explored what portion of CNN, GLM, and LN model performance is specific to a particular stimulus distribution (white noise or natural scenes), versus what portion describes characteristics of the retinal response that generalize to another stimulus class. We found that CNNs trained on responses to one stimulus class generalized better to a stimulus distribution that the model was not trained on (Figure 4B). Despite LN models having fewer parameters, they nonetheless underperform larger convolutional neural network models when predicting responses to stimuli not drawn from the training distribution. GLMs faired particularly poorly when generalizing to natural scene responses, likely because changes in mean luminance result in pathological firing rates after the GLM's exponential nonlinearity. Compared to standard models, CNNs provide a more accurate description of sensory responses to natural stimuli even when trained on artificial stimuli (Figure 4B).

## 3.4   Capturing uncertainty of the neural response

In addition to describing the average response to a particular stimulus, an accurate model should also capture the variability about the mean response. Typical noise models assume i.i.d. Poisson noise drawn from a deterministic mean firing rate. However, the variability in retinal spiking is actually sub-Poisson, that is, the variability scales with the mean but then increases sublinearly at higher mean rates [23, 24]. By training models with injected noise [25], we provided a latent noise source in the network that models the unobserved internal variability in the retinal population. Surprisingly, the model learned to shape injected Gaussian noise to qualitatively match the shape of the true retinal noise distribution, increasing with the mean response but growing sublinearly at higher mean rates (Figure 5). Notably, this relationship only arises when noise is injected during optimization–injecting Gaussian noise in a pre-trained network simply produced a linear scaling of the noise variance as a function of the mean.

## 3.5   Feedforward inhibition shapes temporal responses in the model

To understand how a particular model response arises, we visualized the flow of signals through the network. One prominent aspect of the difference between CNN and LN model responses is that CNNs but not LN models captured the precise timing and short duration of firing events. By examining the responses to the internal units of CNNs in time and averaged over space (Figure 6 A-C), we found that in both convolutional layers, different units had either positive or negative responses to the same stimuli, equivalent to excitation and inhibition as found in the retina. Precise timing in CNNs arises by a timed combination of positive and negative responses, analogous to feedforward inhibition that is thought to generate precise timing in the retina [26, 27]. To examine network responses in

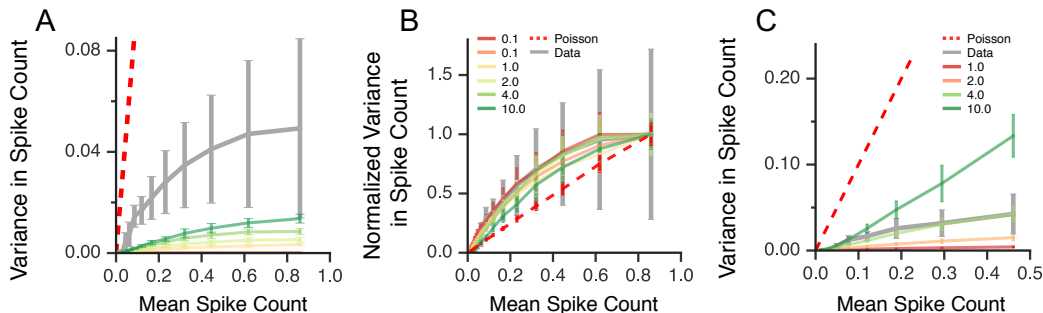

Figure 5: Training with added noise recovers retinal sub-Poisson noise scaling property. (A) Variance versus mean spike count for CNNs with various strengths of injected noise (from 0.1 to 10 standard deviations), as compared to retinal data (black) and a Poisson distribution (dotted red). (B) The same plot as A but with each curve normalized by the maximum variance. (C) Variance versus mean spike count for CNN models with noise injection at test time but *not* during training.

space, we selected a particular time in the experiment and visualized the activation maps in the first (purple) and second (green) convolutional layers (Figure 6D). A given image is shown decomposed through multiple parallel channels in this manner. Finally, Figure 6E highlights how the temporal autocorrelation in the signals at different layers varies. There is a progressive sharpening of the response, such that by the time it reaches the model output the predicted responses are able to mimic the statistics of the real firing events (Figure 6C).

### 3.6   Feedback over long timescales

Retinal dynamics are known to exceed the duration of the filters that we used (400 ms). In particular, changes in stimulus statistics such as luminance, contrast and spatio-temporal correlations can generate adaptation lasting seconds to tens of seconds [5, 28, 14]. Therefore, we additionally explored adding feedback over longer timescales to the convolutional network.

To do this, we added a recurrent neural network (RNN) layer with a history of 10s after the fully connected layer prior to the output layer. We experimented with different recurrent architectures (LSTMs [29], GRUs [30], and MUTs [31]) and found that they all had similar performance to the CNN at predicting natural scene responses. Despite the similar performance, we found that the recurrent network learned to adapt its response over the timescale of a few seconds in response to step changes in stimulus contrast (Figure 7). This suggests that RNNs are a promising way forward to capture dynamical processes such as adaptation over longer timescales in an unbiased, data-driven manner.

## 4   Discussion

In the retina, simple models of retinal responses to spatiotemporal white noise have greatly influenced our understanding of early sensory function. However, surprisingly few studies have addressed whether or not these simple models can capture responses to natural stimuli. Our work applies models with rich computational capacity to bear on the problem of understanding natural scene responses. We find that convolutional neural network (CNN) models, sometimes augmented with lateral recurrent connections, well exceed the performance of other standard retinal models including LN and GLMs. In addition, CNNs are better at generalizing both to held-out stimuli and to entirely different stimulus classes, indicating that they are learning general features of the retinal response. Moreover, CNNs capture several key features about retinal responses to natural stimuli where LN models fail. In particular, they capture: (1) the temporal precision of firing events despite employing filters with slower temporal frequencies, (2) adaptive responses during changing stimulus statistics, and (3) biologically realistic sub-Poisson variability in retinal responses. In this fashion, this work provides the first application of deep learning to understanding early sensory systems under natural conditions.

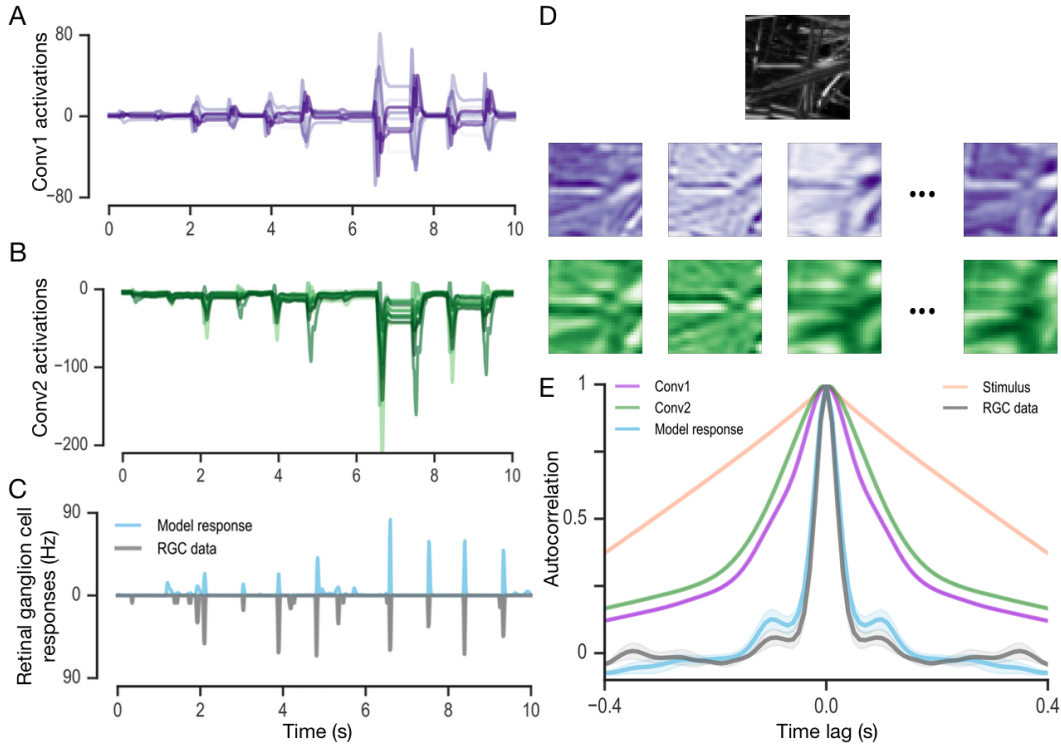

Figure 6: Visualizing the internal activity of a CNN in response to a natural scene stimulus. (A-C) Time series of the CNN activity (averaged over space) for the first convolutional layer (8 units, A), the second convolutional layer (16 units, B), and the final predicted response for an example cell (C, cyan trace). The recorded (true) response is shown below the model prediction (C, gray trace) for comparison. (D) Spatial activation of example CNN filters at a particular time point. The selected stimulus frame (top, grayscale) is represented by parallel pathways encoding spatial information in the first (purple) and second (green) convolutional layers (a subset of the activation maps is shown for brevity). (E) Autocorrelation of the temporal activity in (A-C). The correlation in the recorded firing rates is shown in gray.

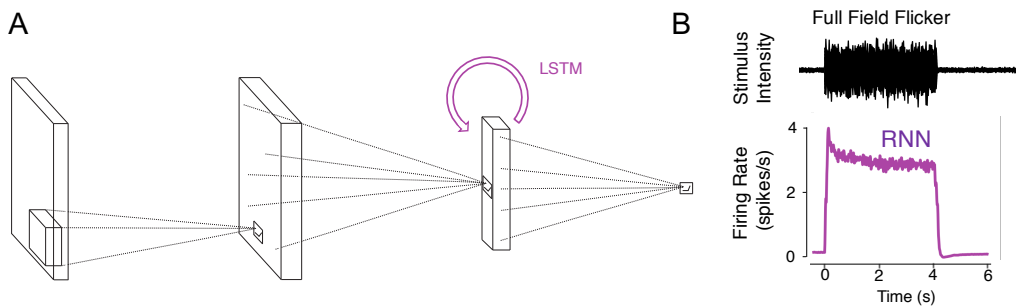

Figure 7: Recurrent neural network (RNN) layers capture response features occurring over multiple seconds. (A) A schematic of how the architecture from Figure 2.1 was modified to incorporate the RNN at the last layer of the CNN. (B) Response of an RNN trained on natural scenes, showing a slowly adapting firing rate in response to a step change in contrast.

To date, modeling efforts in sensory neuroscience have been most useful in the context of carefully designed parametric stimuli, chosen to illuminate a computation or mechanism of interest [32]. In part, this is due to the complexities of using generic natural stimuli. It is both difficult to describe the distribution of natural stimuli mathematically (unlike white or pink noise), and difficult to fit models to stimuli with non-stationary statistics when those statistics influence response properties.

We believe the approach taken in this paper provides a way forward for understanding general natural scene responses. We leverage the computational power and flexibility of CNNs to provide us with a tractable, accurate model that we can then dissect, probe, and analyze to understand what that model captures about the retinal response. This strategy of casting a wide computational net to capture neural circuit function and then constraining it to better understand that function will likely be useful in a variety of neural systems in response to many complex stimuli.

### Acknowledgments

The authors would like to thank Ben Poole and EJ Chichilnisky for helpful discussions related to this work. Thanks also goes to the following institutions for providing funding and hardware grants, LM: NSF, NVIDIA Titan X Award, NM: NSF, AN and SB: NEI grants, SG: Burroughs Wellcome, Sloan, McKnight, Simons, James S. McDonnell Foundations and the ONR.

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
