[Supplementary Material]

# Supplementary Material

## Deep Learning Models of the Retinal Response to Natural Scenes

**Lane T. McIntosh**[*1], **Niru Maheswaranathan**[*1], **Aran Nayebi**[1],
**Surya Ganguli**[2,3], **Stephen A. Baccus**[3]
[1]Neurosciences PhD Program, [2]Department of Applied Physics, [3]Neurobiology Department
Stanford University
{lmcintosh, nirum, anayebi, sganguli, baccus}@stanford.edu

## 1 Supplemental Methods

### 1.1 Retinal recordings

The responses of tiger salamander retinal ganglion cells from 3 animals were recorded using a 60 channel multielectrode array. Further experimental details are described in detail elsewhere [1].

We analyzed the reliability of all recorded cells over the course of each experiment by computing this correlation coefficient between a cell's average response to the same stimulus on different blocks of trials and analyzed only those cells with a correlation exceeding 0.3. Thirty-seven cells exceeded the criterion for reliability. Of these, 70.3% were fast OFF-type cells, 10.8% were medium OFF, and 16.7% were slow OFF. Our original dataset also included ON cells, however none of them passed our retinal reliability criterion.

We interleaved natural scenes and white noise stimuli to average over any experimental drift. However, these transitions generated contrast adaptation over tens of seconds [2, 3] that could not be captured by the short duration of spatiotemporal filters (400 ms) in the CNN. Therefore, we focused our analysis on steady state responses by excluding one minute of data after each transition. Spiking responses were binned using 10 ms bins and smoothed using a 10 ms Gaussian filter.

The training dataset was divided randomly according to a 90%/10% train/validation split, and the test set consisted of averaged repeated trials to 1 minute of novel stimuli.

### 1.2 Stimulus

The white noise stimulus consisted of binary checkers at 35% contrast, and the natural scene stimulus was a sequence of jittered natural images sampled from a natural image database [4].

Of particular note is the spatial resolution of this dataset, which is considerably higher than stimuli used in pre-existing attempts to model retinal responses. Our stimuli consisted of 50 x 50 spatial checkers, each of which spanned 55 $\mu m$ x 55 $\mu m$ on the retina. At this resolution, they can differentially activate nonlinear subunits [5, 6] within the $\sim$250 $\mu m$ salamander ganglion cell receptive field center. Previous stimuli are 120 $\mu m$ pixels, covering roughly the entire RF center in primate peripheral retina [7] or spatially uniform [8, 9]. Since coarser stimuli will not differentially activate subunits, this higher resolution dataset provides a unique challenge for capturing nonlinear retinal responses.

---

[*]These authors contributed equally to this work.

## 1.3 Comparison with other models

To compare the performance of other models, we fit GLMs using spatiotemporal stimulus filters and temporal spike history filters, and LN models using spatiotemporal filters and a parameterized soft rectifying nonlinearity. We found that we needed to regularize the LN model parameters in order to prevent overfitting and offset the large number of parameters (Figure 4A). We tried (1) using a convolutional filter instead of a fully-connected one, (2) only using stimuli centered around the receptive field, and (3) using various levels of $\ell_1$ and/or $\ell_2$ penalties on the filter coefficients as regularization techniques. Cutting out the stimulus around the receptive field (using a window size of 11 x 11 checkers, or 605 $\mu m$), in combination with $\ell_2$ regularization, led to the best held-out performance (Figure 4A). We found the same to be true of GLMs. This cropping regularization procedure resulted in LN models requiring only 4843 parameters, and GLMs requiring 4861 parameters. Therefore, we report LN and GLM performance results using this regularization scheme.

The GLM parameters consist of weights, biases, and spike-history filters, however we did not include cell coupling filters, since [7] reported that adding coupling did not improve predictions of the average firing rate, although they improved single trial predictions. In this study we report performance as the comparison between the models' predictions and averaged responses to repeated stimuli, thus [7] suggests coupling filters would not improve the performance of GLMs.