[Reviews · NeurIPS 2016]

Reviewer 1

Summary

This interesting study models the response of ganglion cells in the retina of a salamander. To the best of my knowledge this is the first time a convolutional neural network is applied to this fairly well studied modeling problem. The authors report better than state of the art performance. Furthermore, and perhaps most interesting in my humble opinion, they report that these models generalize relatively well between artificial white noise stimuli and natural stimuli. This has been a glaring weakness of previous models. Thus I find these results novel and exciting.

Qualitative Assessment

Modeling studies of neural responses are usually measured on two scales: a. Their contribution to our understanding of the neural physiology, architecture or any other biological aspect. b. Model accuracy, where the aim is to provide a model which is better than the state of the art. To the best of my understanding, this study mostly focuses on the latter, i.e. provide a better than state of the art model. If I am misunderstanding, then it would probably be important to stress the biological insights gained from the study. Yet if indeed modeling accuracy is the focus, it's important to provide a fair comparison to the state of the art, and I see a few caveats in that regard: 1. The authors mention the GLM model of Pillow et al. which is pretty much state of the art, but a central point in that paper was that coupling filters between neurons are very important for the accuracy of the model. These coupling filters are omitted here which makes the comparison slightly unfair. I would strongly suggest comparing to a GLM with coupling filters. Furthermore, I suggest presenting data (like correlation coefficients) from previous studies to make sure the comparison is fair and in line with previous literature. 2. The authors note that the LN model needed regularization, but then they apply regularization (in the form of a cropped stimulus) to both LN models and GLMs. To the best of my recollection the GLM presented by pillow et al. did not crop the image but used L1 regularization for the filters and a low rank approximation to the spatial filter. To make the comparison as fair as possible I think it is important to try to reproduce the main features of previous models. Minor notes: 1. Please define the dashed lines in fig. 2A-B and 4B. 2. Why is the training correlation increasing with the amount of training data for the cutout LN model (fig. 4A)? 3. I think figure 6C is a bit awkward, it implies negative rates, which is not the case, I would suggest using a second y-axis or another visualization which is more physically accurate. 4. Please clarify how the model in fig. 7 was trained. Was it on full field flicker stimulus changing contrast with a fixed cycle? If the duration of the cycle changes (shortens, since as the authors mention the model cannot handle longer time scales), will the time scale of adaptation shorten as reported in e.g Smirnakis et al. Nature 1997.

Confidence in this Review

3-Expert (read the paper in detail, know the area, quite certain of my opinion)


Reviewer 2

Summary

This paper applied convolution neural networks (CNNs) to the responses of retinal ganglion cells to white noise and natural scenes. The conclusions are that optimized CNN models provide a more accurate description of neural responses compared to previous works that used simpler linear-nonlinear (LN) models. Unfortunately, there are major problems with how LN models were implemented for comparisons with CNNS, and with the interpretation of optimized CNNs themselves. As a result, the work appears to contradict a large body of neurophysiological studies on the retina.

Qualitative Assessment

There are major problems with the study design and interpretation. The existing literature on retinal physiology and modeling was not adequately cited and discussed. Some of the citations are included above in this report. Major conclusions of this paper, such as "convolutions neural network models markedly outperform previous models in predicting retinal responses" are not justified.

Confidence in this Review

3-Expert (read the paper in detail, know the area, quite certain of my opinion)


Reviewer 3

Summary

The authors demonstrate that deep convolutional neural networks (CNNs) capture retinal responses to natural scenes nearly to within the variability of a cell’s response, and are more accurate than linear-nonlinear (LN) models. They find two additional properties of CNNs: they are less susceptible to overfitting than their LN counterparts when trained on small amounts of data, and generalize better when tested on stimuli drawn from a different distribution (e.g. between natural scenes and white noise). It is an interesting paper and as far as I know, the first paper to apply deep learning models to retinal data. I also like how they probed further the activity of the internal units to get a better understanding of the mechanism underlying the generation of temporally precise responses.

Qualitative Assessment

Major: 1) There has been a fair amount of work using variations of LN cascade models, which have been shown to perform better than LN models. Did the authors try to compare the performance of their model against such cascade LN models? In particular, it would be interesting to see how this work compares to the convolutional sub-unit model used for responses in primate V1 (Vintch et al, NIPS 2012 and Vintch et al, J. Neuroscience 2015). How similar/different is the authors’ model from Vintch et al? Minor: 1) Line 73 should be ‘ ….. Fig 1’. 2) Line 66: The dataset consists only of OFF cells. Could the authors speculate on if and how results might change with inclusion of ON cells? 3) Line 83: Why filter sizes in excess of 15 x 15 checkers? Is this correlated with typical RF sizes in salamander retina? 4) Line 128: I am not sure I understand the first sentence. Are the authors able to delineate specific aspects of the CNNs that allows them to generalize to another stimulus class, compared with LN models? If so, I would request that this be clarified. 5) Figure 6: Please clarify in Panel C that RGC data firing rates are not actually negative. Please clarify what the shading in Panel E represents. 6) Figure 7: Is it possible to estimate the timescale of firing rate adaption in panel B? How does this compare to biological estimates? 7) In figures, do error bars represent standard deviations or standard error of mean? Please clarify.

Confidence in this Review

3-Expert (read the paper in detail, know the area, quite certain of my opinion)


Reviewer 4

Summary

This paper fits CNNs to the joint responses of RGCs to naturalistic and artificial stimuli. The authors compare the prediction performance of the CNNs to standard models from the field and analyze the properties of the fit network. The noise shaping analysis was in particular interesting and novel.

Qualitative Assessment

The paper is well written and interesting and contains some interesting analysis of the fit CNN model. The scientific results regarding what can be learned from the CNN about retinal processing are still preliminary, making this paper an excellent candidate for a poster.

Confidence in this Review

3-Expert (read the paper in detail, know the area, quite certain of my opinion)


Reviewer 5

Summary

This paper introduces Convolutional Neural Networks (CNN) as a computational model to predict the retinal response to natural scenes. The authors trained CNNs on recorded ganglion cells from 3 tiger salamanders and compared the results with standard models for retinal response such as Linear-Nonlinear (LN) models and Generalized Linear Models (GLMs). Results show that CNNs are better predictors for both white noise and natural scenes stimuli. Moreover, CNNs better perform also in explaining the variance of input noise.

Qualitative Assessment

The paper shows promising results in modelling the retinal response with CNNs. However, a substantial lack of details of the implemented networks affects the reproducibility of the results. For example, I could not find an explanation of the spatiotemporal filters (the mentioned 400ms of temporal filtering) used, if and how the dataset was divided into training and validation sets, the number of epochs used or the final error reached. Similarly, the LN and GLM models details are not explained. In the caption of Figure 4 it is said that there are 150k parameters for the CNN and 100k parameters for LN – no mention to GLM – thus some questions arise: could be the case that the CNN performance are due to the larger number of parameters involved? Should not be the model performance scaled by the number of parameters used (such as a Bayesian information criterion, for example)? What are, in the authors’ opinions, the reasons of such a fail for the LN and GLM models? Finally, section 3.6 just introduces an interesting part of the work about Spike Frequency Adaptation (SFA), but it is still lacking of the full implementation details. This part seems rather preliminary. SFA has been modeled using power-law kernels (see Pozzorini, C., Naud, R., Mensi, S., Gerstner, W. (2013). Temporal whitening by power-law adaptation in neocortical neurons, Nature Neuroscience; or Bohte, S. M. (2012). Efficient Spike-Coding with Multiplicative Adaptation in a Spike Response Model. NIPS 2012). Thus, would have been more significant to compare the obtained results with such models.

Confidence in this Review

2-Confident (read it all; understood it all reasonably well)


Reviewer 6

Summary

This paper fits a convolutional neural network to the outputs of retinal ganglion cells. The dataset includes input-output pairs where the input is a controlled image, such as a natural image or a white noise image, and the output is the corresponding recordings from retinal ganglion cells. The paper demonstrates that convolutional networks provide better fits than generalized linear models or linear-nonlinear models.

Qualitative Assessment

The paper shows that convolutional networks provide better fits to retinal response than GLMs and linear-nonlinear models. While this result is novel, in my opinion, it is widely expected. It neither provides algorithmic progress, nor neuroscientific progress. I would be much more excited if the layers of the neural network were somehow mapped to the layers of the retinal processing pipeline (i.e., horizontal cells, bipolars, amacrines.). The fact that the accuracy did not improve beyond 3 layers is not analyzed at all, which I think could be interesting from a retinal processing point of view. Similarly, the rather large filter sizes are not analyzed either.

Confidence in this Review

3-Expert (read the paper in detail, know the area, quite certain of my opinion)